# Vaccine versus Variants (3Vs): Are the COVID-19 Vaccines Effective against the Variants? A Systematic Review

**DOI:** 10.3390/vaccines9111305

**Published:** 2021-11-10

**Authors:** Kadhim Hayawi, Sakib Shahriar, Mohamed Adel Serhani, Hany Alashwal, Mohammad M. Masud

**Affiliations:** 1College of Technological Innovation, Zayed University, Abu Dhabi 51133, United Arab Emirates; Abdul.Hayawi@zu.ac.ae (K.H.); sakib.shahriar@zu.ac.ae (S.S.); 2College of Information Technology, UAE University, Abu Dhabi 15551, United Arab Emirates; halashwal@uaeu.ac.ae (H.A.); m.masud@uaeu.ac.ae (M.M.M.)

**Keywords:** SARS-CoV-2, variants of concern, vaccine effectiveness, COVID-19, vaccine efficacy

## Abstract

Background: With the emergence and spread of new SARS-CoV-2 variants, concerns are raised about the effectiveness of the existing vaccines to protect against these new variants. Although many vaccines were found to be highly effective against the reference COVID-19 strain, the same level of protection may not be found against mutation strains. The objective of this study is to systematically review relevant studies in the literature and compare the efficacy of COVID-19 vaccines against new variants. Methods: We conducted a systematic review of research published in Scopus, PubMed, and Google Scholar until 30 August 2021. Studies including clinical trials, prospective cohorts, retrospective cohorts, and test negative case-controls that reported vaccine effectiveness against any COVID-19 variants were considered. PRISMA recommendations were adopted for screening, eligibility, and inclusion. Results: 129 unique studies were reviewed by the search criteria, of which 35 met the inclusion criteria. These comprised of 13 test negative case-control studies, 6 Phase 1–3 clinical trials, and 16 observational studies. The study location, type, vaccines used, variants considered, and reported efficacies were highlighted. Conclusion: Full vaccination (two doses) offers strong protection against Alpha (B.1.1.7) with 13 out of 15 studies reporting more than 84% efficacy. The results are not conclusive against the Beta (B.1.351) variant for fully vaccinated individuals with 4 out of 7 studies reporting efficacies between 22 and 60% and 3 out of 7 studies reporting efficacies between 75 and 100%. Protection against Gamma (P.1) variant was lower than 50% according to two studies in fully vaccinated individuals. The data on Delta (B.1.617.2) variant is limited but indicates lower protection compared to other variants.

## 1. Introduction

As of 31 August 2021, 217 million infections and more than 4.5 million deaths [1] have been attributed to SARS-CoV-2, commonly referred to as the COVID-19 pandemic. The worldwide impacts on sectors including the economy [2] and public health [3] were catastrophic due to public restrictions and lockdown. A potential solution for ending the pandemic was introduced by the emergence of safe and effective vaccines [4]. Early clinical trials for many vaccines were reported to be highly effective against the reference SARS-CoV-2 variant [4,5] and more than 5 billion vaccine doses have been administered by 31 August 2021 [1]. However, once new variants of the virus were discovered in the population, there has been a growing concern regarding the level of protection ensured by vaccines. A mathematical modeling demonstrated the possibility of acquiring a high level of immunity in the population if the vaccine efficacy is maintained across the different strains [6]. The level of vaccine efficacy required to reach a communal level of protection poses a challenging question. According to computational modeling and simulation experiments, a minimum of 60% vaccine efficacy is required to end an ongoing epidemic if the vaccination coverage is 100% [7]. However, given that a 100% vaccine coverage is highly unlikely due to various reasons including vaccine hesitancy and shortages in many parts of the world, the required efficacy increases further. The same study simulates that 80% efficacy is required for a vaccine coverage of 75% to end an ongoing epidemic. According to another analysis, efficacy of 50% or greater could substantially reduce incidence of COVID-19 in vaccinated individuals and would offer useful herd immunity [8]. Therefore, it is essential to find out if the current vaccines can offer the desired level of efficacies amid the increasing transmission of newer SARS-CoV-2 variants of concerns.

In many parts of the globe, the variants now constitute the majority of the COVID-19 cases [9]. Figure 1 illustrates the four variants of concerns in the United States (US) along with their first detection location, level of spread, and effects using data from the Centers for Disease Control and Prevention (CDC) [10].

Viruses constantly evolve by mutations and therefore the detection of SARS-CoV-2 variants are not unexpected. However, the appearance of these new and potentially more infective SARS-CoV-2 variants can make it more challenging to end the global pandemic [11,12]. The B.1.617.2 (Delta) and B.1.1.7 (Alpha) variants are found to be more infectious and can lead to more community transmission. The increase in viral transmission can consequently result in the emergence of more variants. Large-scale vaccine deployment is considered an important tool against these variants [11]. As such, it is essential to know the effectiveness of the COVID-19 vaccines especially against the new variants of concerns. The lack of data on the efficacy of vaccines against the new variants creates an uncertainty that can prolong the pandemic [13]. Therefore, the objective of this research is to systematically review the most recent literature on the effectiveness of the vaccines on the variants of concerns. We are hopeful that this research will help the public health community in understanding which variants are more resistant to vaccines and consequently help in new vaccine development efforts. The rest of the paper is organized as follows: Section 2 presents the systematic review search strategy and study inclusion criteria. The vaccine efficacy results against the different variants are discussed and tabulated in Section 3. In Section 4, the efficacy comparison between the reference variant and the new variants as well as the protection for older aged population against the new variants are discussed. Section 5 concludes the paper and summarizes the key findings. 

## 2. Materials and Methods

The review followed the appropriate guidelines specified in the Preferred Reporting Items for Systematic Reviews and Meta-Analyses (PRISMA) statement [14]. The search strategy and results are presented next.

### 2.1. Search Strategy

Scopus and PubMed databases were searched for studies reporting vaccine effectiveness against SARS-CoV-2 variants using the following search criteria: (COVID-19 OR SARS-CoV-2) AND (“Vaccine effective*” OR “Vaccine efficacy”) AND “Variant*”. The search criteria were applied to the title, abstract, and keywords. Moreover, only articles published in 2020 and 2021 in the English language were included in the criteria. No restrictions in terms of the study design and geographical region were applied in the search. Google scholar was also searched, and search results have been analyzed to select only relevant gray pieces of literature, latest published articles not updated on databases, and preprints. The last Google Scholar search was performed on 31st of August 2021. 

### 2.2. Systematic Review Results

The review selection process according to the PRISMA guidelines is illustrated in Figure 2. A total of 192 articles were identified through database searching and 129 of them were screened after removing duplicates. In the screening process, 63 non-relevant articles, including discussions of vaccination strategies and non-variant articles, were removed after reviewing their titles and abstracts. Out of 66 articles accessed for full-text, 35 were included for the final study. A total of 12 (38.7%) of the excluded articles focused on serum neutralization effects on variants, 7 (22.6%) conducted non-human studies, and 8 (25.8%) were either review papers or analyses. A further four of the excluded articles discussed vaccine strategies, immunogenic comparison, and variant detection techniques. 

### 2.3. Risk of Bias Assessment

The included studies were evaluated using the risk of bias in the prevalence tool with a resulting estimate in line with Cochrane GRADE criteria of low, moderate, or high risk of bias [15]. A series of 10 questions related to the sampling and data collection of the studies were answered using this tool. Each question is answered with a yes/no response and a lack of information is assumed to be unclear/no. The final score is a sum of all negative answers and scores of 0–3 are considered low risk, 4–6 are moderate risk, and 7–10 are high risk. 

## 3. Results

The characteristics of the included studies are summarized next. Out of the 35 included studies, 20 (57%) were peer-reviewed and 15 (43%) were preprints or reports. The Delta (B.1.617.2) variant was the latest variant to be declared a variant of concern (VOC) by Public Health England on 6th May 2021 [9]. Therefore, the majority (70%) of the studies reporting efficacy against the Delta variant were not peer-reviewed at the time of writing this paper. However, these research works are still expected to provide an early indication of the vaccine efficacy against the Delta variant. In terms of location, eight studies were from the United Kingdom (UK), seven from the United States (US), five from South Africa, four from Canada, three from Qatar, two each from France and Brazil, one each from Italy, India, Israel, and Europe. 

According to the risk of bias tool for assessing the included studies, three of the studies were considered to be at a high risk of bias, 15 were considered to be at a moderate risk of bias, and 17 were considered to be at a low risk of bias. The results are summarized in Table 1, and the full scoring details are provided in the Appendix A. 

The following subsections discusses the vaccine effectiveness and provides a comparison against the newer variants.

### 3.1. Vaccine Protection against Alpha (B.1.1.7) SARS-CoV-2 Variant

Given that B.1.1.7 was the earliest variant to be designated a VOC, more data of vaccine efficacy is available for this variant than any other. A total of 21 of the 35 (60%) studies reported vaccine efficacy against this variant, as summarized in Table 2. 

Several articles reported efficacies from Phase 2 and 3 clinical trials ([16,17,18]). The efficacies range from 70.4% and 85.6% against B.1.1.7 for AstraZeneca and Novavax vaccines. The Novavax Phase 3 trial was conducted between 28 September and 28 November 2020, on 16,645 adults aged between 18 and 84 years in the UK [16]. A Post hoc analysis showed that the efficacy against the Alpha (B.1.1.7) variant was lower than the non-B.1.1.7 variants, the majority of which are likely to be the reference strain given the study period. The efficacy against the Alpha variant was 86% whereas the efficacy against the non-Alpha variants were higher at 96.4%. The results from this clinical trial are further confirmed by E. Mahase [18]. A Phase 2 trial for the AstraZeneca vaccine [17] with 8534 adults aged 18 or older from 31 May and 13 November 2020, in the UK also confirmed a lower efficacy for B.1.1.7 through post hoc analysis. A vaccine efficacy of 70.4% was reported against the Alpha variant whereas the efficacy was higher at 81.5% for non-Alpha variants. 

From the observational studies, a wider range of efficacies are reported. The efficacy of a single dose vaccine was reported to be as low as 29.5% for BNT162b (Pfizer–BioNTech) [19] and as high as 88.1% for mRNA-1273 (Moderna) [20]. Similar variability is observed for two doses of vaccines where the lowest efficacy reported was 74.5% for two doses of ChAdOx1 nCoV-19 (AstraZeneca) vaccine [21] and the highest efficacy of 100% for mRNA-1273 (Moderna) [20]. Efficacy of 50% or greater is substantial and would offer useful herd immunity [8]. Overall, the results in Table 2 indicate that two doses of any reported vaccines provide good protection against B.1.1.7. Full vaccination (two doses) offers strong protection against Alpha with 13 out of 15 studies reporting more than 84% efficacy and a single dose offers reasonable protection with 10 out of 12 studies reporting greater than 54% efficacy. Moreover, the efficacies reported by the mRNA vaccines (BNT162b and mRNA-1273) appear to be slightly higher comparatively. Furthermore, protection against B.1.1.7 is estimated to be higher than B.1.617.2 [22]. An analysis of 76 breakthrough cases after full vaccination in metropolitan New York was presented by R. Duerr et al. [23]. It was observed that 4 of the 7 hospitalized cases were infected with the Alpha variant, including 1 death. The study also reported that the vaccines offer strong protection against B.1.1.7 statistically but the level of efficacy was not quantified. 

### 3.2. Vaccine Protection against Beta (B.1.351) and Gamma (P.1) SARS-CoV-2 Variant

Both E484K-positive mutations, B.1.351 (E484K and K417N) and P.1 (E484K and K417T) were reported together in several studies. The results against these two variants are summarized in Table 3. 

Several phases 1–3 clinical trials reported efficacies against the Beta (B.1.351) variant ([18,37,38,39]). The lowest efficacy reported is 49.4% with at least one dose of NVX-CoV2373 [37] and the highest efficacy is 100% for two doses of BNT162b2 [39]. The NVX-CoV2373 Phase 2A,B randomized, observer-blinded controlled trial was conducted in South Africa between 17 August and 25 November 2020 [37]. The overall vaccine efficacy was 49.4% but was slightly higher at 51.0% in HIV-negative participants. No serious adverse effects were reported as being related to the vaccine. E. Mahase [18] reported an efficacy of 60% for the NVX-CoV2373 trial against the B.1.351 variant. A Phase 3 randomized, double-blind, placebo-controlled trial for a single dose of Ad26.COV2.S vaccine was conducted in several countries including US, Brazil, and South Africa [38]. The trial period was between 21 September 2020, and 22 January 2021. The study included 4969 participants from South Africa, where the B.1.351 variant was dominant. The vaccine efficacy from the South Africa group was 52.0% for moderate to severe-critical COVID-19 after 14 days of administration and increased to 64.0% after 28 days. Serious adverse effects not related to COVID-19 were only reported by 83 out of 21,895 vaccine recipients constituting 0.4% of the population and 96 out of 21,888 placebo recipients constituting 0.4% of the population. A Phase 2 observer-blinded controlled trial for BNT162b2 was conducted between July and October 2020 and included 45,411 participants aged 16 or older from US, Argentina, Brazil, South Africa, Germany, and Turkey [39]. Although a 91% overall efficacy was reported, the efficacy specifically for the South African group with predominantly B.1.351 variant in circulation was reported to be 100%. The paper also reported new adverse events attributed to the BNT162b2 vaccine not identified in earlier reports. These adverse events decreased appetite, lethargy, asthenia, malaise, night sweats, and hyperhidrosis. The study however has not been peer-reviewed at the time of writing. Moreover, for fully vaccinated individuals, 4 out of the 7 studies reported efficacies between 22 and 60%, and 3 reported efficacies between 75 and 100%. Therefore, more data are needed for convincing evidence. 

The efficacies reported against Gamma (P.1) variant is all from observational studies. Three of the involved studies reported efficacies for the P.1 variant exclusively ([35,40,41]). Protection against this variant is indicated to be lower ranging from 12.5% for a single dose of CoronaVac [40] and 61% for a single dose of mRNA vaccine (BNT162b2 and mRNA-1273) [35]. Both studies on the CoronaVac vaccine against the P.1 variant were conducted in Brazil ([40,41]). The test negative case-control study [40] included 43,744 adults aged 70 years or older between 17 January and 29 April 2021. For two doses of CoronaVac, the vaccine efficacy against COVID-19 infection was only at 46.8%. However, the adjusted effectiveness against COVID-19 hospital admissions and deaths was 55% and 61.2% respectively. The study by M. D. T. Hitchings et al. [41] meanwhile included 106,329 healthcare workers aged 18 years or older between 19 January and 25 March 2021. The adjusted vaccine effectiveness was 49.6% after the first dose of CoronaVac. However, the effectiveness reported after two doses was unusually lower at 36.8%. This study was not peer-reviewed at the time of writing. Overall, the limited data available against the P.1 variant indicates lower protectiveness. The highest efficacy was only 61% with a single dose of mRNA vaccine but can be expected to be higher with full vaccination. 

For studies reporting efficacies of both B.1.351 and P.1 variants together ([32,33,36]), the reported efficacies for double dose mRNA vaccines are 84%, 88%, and 77% respectively. The high efficacies against both variants combined seem to indicate slightly greater protection against B.1.351 compared to P.1. This conclusion can be drawn solely based on the efficacy results available exclusively against the two variants where the range of efficacy against the B.1.351 variant was between 22% and 100% and the range of efficacy against the P.1 variant was between 12.5% and 61%. 

### 3.3. Vaccine Protection against Delta (B.1.617.2) SARS-CoV-2 Variant

Among the variants considered, the Delta (B.1.617.2) variant is the latest to be designated as a VOC. Consequently, only 30% of the research reporting vaccine efficacies against this variant is from peer-reviewed literature and no data from a clinical trial is available at the time of writing. The studies reporting vaccine effectiveness against the B.1.617.2 variant are summarized in Table 4. 

Several studies have indicated a drop in protection against the Delta variant. Vaccine efficacy for Mesa County, Colorado was reported to be 78% with Delta being the prevalent variant [44]. In contrast, for the same period of time, efficacy from the other Colorado counties was 89% where the Delta variant was comparatively lower. Similarly, S. Y. Tartof et al. [45] indicated lower protection against Delta (75% efficacy) compared to other variants (91% efficacy). A sharp decline in vaccine effectiveness is reported after the Delta variant became prevalent with the effectiveness reducing from 91% to 66% [46]. This is also indicated by A. Puranik et al. [22] where a sharp decline in efficacy is observed from July 2021 when the Delta variant replaced the Alpha variant as the dominant variant (Figure 3). 

Moreover, among the two mRNA vaccines, several studies indicate slightly higher protection from the Moderna (mRNA-1273) vaccine compared to the Pfizer (BNT162b2) vaccine. For both single and double doses, mRNA-1273 (79% and 84.8%) offered greater protection than BNT162b2 (64.2% and 53.5%) against the Delta variant [47]. Similarly, for two doses of either, mRNA-1273 (76%) was reported to be more effective than BNT162b2 (42%) [22]. For a single dose of mRNA vaccine, a higher efficacy for mRNA-1273 (72%) than BNT162b2 (56%) is reported against the Delta variant [36]. 

## 4. Discussion

The effectiveness of COVID-19 vaccines against the new variants was highlighted and compared in the previous sections. Given that vaccine effectiveness of higher than 50% could substantially reduce incidence of COVID-19 in vaccinated individuals [8], the existing vaccines from the available data indicates a good level of protection against the newer variants, especially with two doses. The efficacy against the Alpha variant is reported to be the highest, followed by Beta and Gamma variants. However, from the limited data available, the efficacy against Delta variant is indicated to be lower but still effective. Based on the studies, relevant discussions and comparison is presented hereafter. 

### 4.1. Protection against COVID-19 Reference Variant and New Variants

Lower vaccine effectiveness against the newer variants is expected given the mutation. NVX-CoV2373 [18] vaccine was found to be 95.6% effective against the reference variant compared to only 85.6% and 60% for Alpha and Beta variants, respectively. Similarly, a drop in efficacy against an Alpha variant of 86.3% compared to 96.4% against non-Alpha variants, the majority of which was the reference variant was observed [16]. An observational study from France indicated slightly lower protection for Alpha (86%) and Beta/Gamma (77%) compared to the reference variant (88%) [32]. Therefore, data from the studies that have reported efficacies both against the new variants as well as the original reference variant confirms the decrease in efficacy with the emergence of variants. This decrease in efficacy is consistent with the influenza virus analyzed data where a reduction in vaccine efficacy against an emerging variant was reported in [51], with the effectiveness for influenza A/H1N1 of 65% being reduced to 39% for the circulating A/H3N2 variant. 

### 4.2. Protection for Older Population against SARS-CoV-2 Variants

The effectiveness of vaccines in older aged population always remains a significant point of discussion. The development of immunologic memory declines with age for primary and booster vaccination [52]. Due to aging, a reduction in naive T cells that respond to a vaccine is observed, with a significant decrease in CD8 T cells. Impaired CD8 T-cell effector responses and reduced CD4 T-cell functionality contributes to lower vaccine response in older population. An earlier study on COVID-19 vaccines in older population reported concern over limited trial data available for this age group [53]. Therefore, research works that have reported vaccine efficacies against the newer variants with an exclusively older age population in their studies are presented next. In a test negative case-control study [25] during Alpha variant prevalence, both BNT162b2 and ChAdOx1 nCoV-19 vaccines were found to be effective. More significantly, a single dose of either vaccine was found to be around 80% effective in preventing COVID-19-related hospital admissions for the older population. The effectiveness of one dose mRNA vaccine among the older population was investigated by D. M. Skowronski et al. [35]. The efficacies after the first dose were 65% overall, 67% against Alpha, and 61% against Gamma for both mRNA vaccines aggregated. Moreover, adults aged 70 or older from Brazil participated in a test negative case-control study where the efficacy of the CoronaVac vaccine was 55.5% against COVID-19 hospital admissions and 61.2% against COVID-19 deaths against the Gamma variant [40]. Similarly, the test negative case-control study [31] involved participants aged 65 or older across Europe during predominantly Alpha variant period in which for fully vaccinated BNT162b2 individuals, the protection against Alpha was high (87%) and for one dose of ChAdOx1 nCoV-19 the protection was reasonable (68%). Finally, a prospective cohort study involving care home residents aged 65 or older in England was conducted during the prevalence of Alpha variant [29]. The reported effectiveness was 56% (28 to 34 days) and 62% (35 to 48 days) after a single dose of ChAdOx1 or BNT162b2. Therefore, from the studies that exclusively targeted an older population, the vaccines appear to offer reasonable protection against the variants especially against severe illness and deaths. 

### 4.3. Protection against Other SARS-CoV-2 Variants

Besides the four major VOCs considered in this review, an observational study [23] reported vaccine protection against the B.1.526 (Iota) variant. The study looked at 76 breakthrough cases of individuals who received full vaccination of BNT162b2, mRNA-1273, and Ad26.COV2.S between February and April 2021. Although statistical analysis concluded an effective level of protection against B.1.526, the study did not quantify the vaccine efficacy. The efficacy against non-VOC was found to be higher (72%) compared to the Alpha (67%) and Gamma (61%) VOCs [35]. However, the exact variants present in the non-VOC group was not reported. Overall, from the existing works in the literature, only limited data are available against other variants and non-VOCs in particular. However, it must be noted that this is most likely due to the dominance of the four discussed variants in the current study period. 

### 4.4. Significance and Implication of This Review

This systematic review looked at existing research and compared the efficacies of vaccines against SARS-CoV 2 variants. Concerns over the vaccine efficacies against the new variants have been emphasized [54]. However, our study demonstrates that the vaccines still provide strong protection against the variants especially in terms of severe illness and hospitalization. Moreover, we are hopeful that our study will support the efforts to increase vaccine acceptance with two doses for vulnerable people including the elderly population and those with comorbidities especially against the Alpha and Beta variants. 

This review provides a compressive and comparative study on vaccine efficacies against various variants of COVID-19 and there are several key implications of this review. First, it helps in increasing public health awareness among the population and addresses their concerns regarding the efficacies of the vaccines against new variants. Moreover, by comparing the differences in efficacies between the various vaccines will help in prioritizing further developments of these vaccines and help public decision making. Furthermore, this type of comparative and comprehensive review assimilating the latest research supports policy makers, governments, and public health community in assessing the stability, sustainability, and resilience of vaccines against emerging variants.

However, there remains some unanswered research questions regarding the efficacies of vaccines against the variants which will lead to the development of future research direction. First, it is still not evident what the best approach is in designing a vaccine that can be tested against and unknown or non-prevalent variant. Researchers should also investigate the possibility of predicting emerging variants and the best vaccine design for protection against these variants. Moreover, design and development of a safe and effective vaccine that can offer strong protection against all existing variants and future mutations should be examined. Recent studies have also indicated that booster vaccinations can be a useful long-term global health strategy if the existing vaccines cannot offer significant protection against emerging variants [55]. Therefore, researchers are encouraged to examine existing vaccine efficacy data from booster vaccines against the variants.

### 4.5. Strengths and Limitations of This Review

The systematic review to the best of our knowledge is the first one addressing the efficacies of vaccines against the SARS-CoV-2 variants. The search strategies and inclusion criteria used was clearly defined and PRISMA recommendations for systematic reviews were followed. Additionally, risk of bias assessment was performed to assess the included research articles. Although most of the included articles were peer-reviewed, many preprints were also included. Preprints were useful for obtaining the most recent results due to the topic being part of a very active area of research. Furthermore, in light of the COVID-19 pandemic, preprints now offer an unprecedented level of research significance [56] and therefore their usage is appropriate.

This review also has some limitations. First, meta-analysis was not performed in this work and the lack of statistical analysis can be considered a weakness. Moreover, different approaches to vaccine efficacy calculation and estimation were used in the included studies and in this review the results were presented without adjusting for the different methods. Finally, Medline and Embase databases were not used leading to a possibility of missing a small number of published studies.

## 5. Conclusions

Studies of vaccine’s effectiveness have been considered a key factor for any vaccine’s evaluation to ensure higher degree of protection. These studies are more relevant for viruses that continuously mutate to new variants were vaccine’s efficacities might decline with new derivations of the virus. Various research has been conducted to access the COVID-19 vaccine’s efficacies. However, no comprehensive study has been conducted to assess and compare efficiency of vaccines with various variants of COVID-19.

This systematic review compared the vaccine efficacies of COVID-19 vaccines against the newer variants. A total of 35 research articles were included in the review and included test negative case-control studies, Phase 1–3 clinical trial results, and observational studies. Results indicate that although efficacy is lower than the reference strain, the vaccines offer a good level of protection against the newer variants, especially with two doses. Protection against the Alpha variant is highest, followed by Beta and Gamma variants. Efficacy seems to be lower for the Delta variant from the limited data available. Results also indicate a good level of protection for the older population aged 60 or older. However, not enough data are available for other variants besides Alpha, Beta, Gamma, and Delta. 

## Figures and Tables

**Figure 1 vaccines-09-01305-f001:**
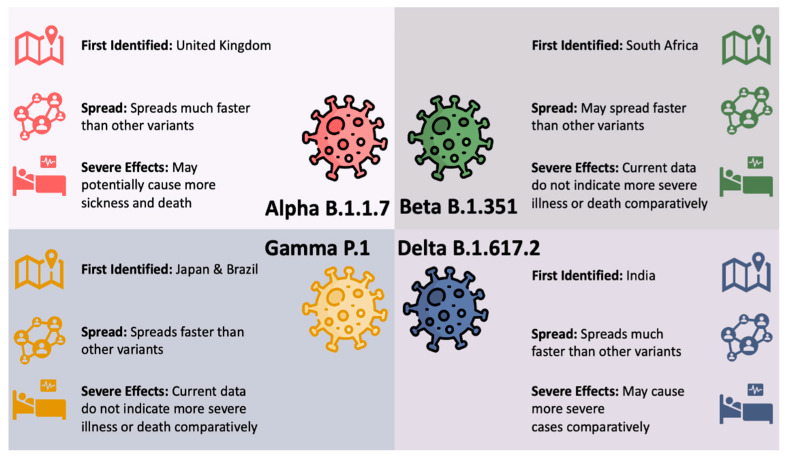
SARS-CoV-2 Variants of Concern in the US.

**Figure 2 vaccines-09-01305-f002:**
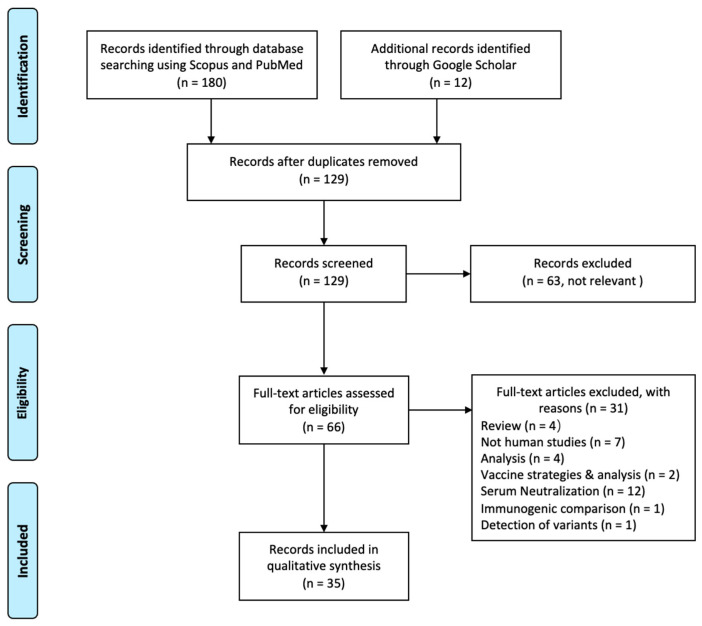
PRISMA Flow Diagram.

**Figure 3 vaccines-09-01305-f003:**
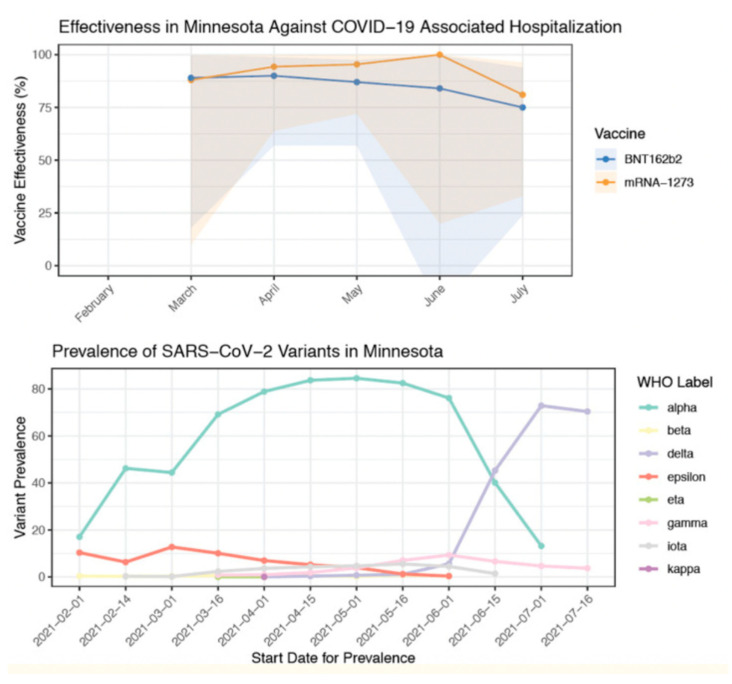
Decrease in Vaccine Efficacy Against Delta (B.1.617.2) Variant [22].

**Table 1 vaccines-09-01305-t001:** Risk of Bias Assessment.

Study	Overall Risk of Bias
P. T. Heath et al. [16]	Low
K. R. W. Emary et al. [17]	Low
E. Mahase [18]	High
L. J. Abu-Raddad et al. [19]	Moderate
H. Chemaitelly et al. [20]	Low
J. Lopez Bernal et al. [21]	Low
A. Puranik et al. [22]	Low
R. Duerr et al. [23]	Moderate
M. W. Tenforde et al. [24]	Moderate
J. Lopez Bernal et al. [25]	Low
E. J. Haas et al. [26]	Low
V. J. Hall et al. [27]	Moderate
M. E. Flacco et al. [28]	Moderate
M. Shrotri et al. [29]	Low
V. J. Hall et al. [30]	Moderate
E. Kissling et al. [31]	Moderate
T. Charmet et al. [32]	Moderate
H. Chung et al. [33]	Moderate
A. Yassi et al. [34]	Moderate
D. M. Skowronski et al. [35]	Low
S. Nasreen et al. [36]	Low
V. Shinde et al. [37]	Low
J. Sadoff et al. [38]	Low
S. J. Thomas et al. [39]	Low
O. T. Ranzani et al. [40]	Low
M. D. T. Hitchings et al. [41]	Moderate
S. A. Madhi et al. [42]	Low
B. Lefèvre et al. [43]	Moderate
R. Herlihy et al. [44]	High
S. Y. Tartof et al. [45]	Low
A. Fowlkes et al. [46]	Moderate
P. Tang et al. [47]	Moderate
X.-N. Li et al. [48]	Low
R. Thiruvengadam et al. [49]	Moderate
L. T. Keegan et al. [50]	High

**Table 2 vaccines-09-01305-t002:** Vaccine Effectiveness Against Alpha (B.1.1.7) Variant.

Reference	Study Type	Population and Period	Location	Peer-Reviewed	Vaccine +	Reported Vaccine Efficacy
[16]	Phase 3, randomized, observer-blinded, placebo-controlled trial	15,187 participants aged 18–84. 28 September–28 November 2020	UK	√	NVX-CoV2373	86.3% (95% CI 71.3–93.5) against B.1.1.7 and 96.4% (73.8–99.5) against non-B.1.1.7 variants
[17]	Single-blind, randomized Phase 2 trial	8534 participants aged ≥18. 31 May–13 November 2020	UK	√	ChAdOx1 nCoV-19	70.4% (95% CI 43.6–84.5) against B.1.1.7 and 81.5% (67.9–89.4) against non-B.1.1.7 variants
[18]	Phase 2 and Phase 3 trials	N/A	UK	Χ	NVX-CoV2373	85.6% (CI Not reported) *****
[19]	Test negative case-control study	Qatari resident. February-March, 2021	Qatar	√	BNT162b2	Single dose: 29.5% (95% CI 22.9–35.5) Double dose: 89.5% (85.9–92.3) ≥14 days after the second dose
[20]	Test negative case-control study	50,068 PCR-confirmed and negative for B.1.1.7 Qatari residents. 28 December 2020–10 May 2021	Qatar	√	mRNA-1273	Single dose: 88.1% (95% CI 83.7–91.5) ≥14 days after first dose Double dose: 100% (91.8–100.0) ≥14 days after the second dose
[21]	Test negative case-control study	19,109 participants aged ≥16. 26 October 2020–30 May 2021	UK	√	BNT162b2, ChAdOx1 nCoV-19	BNT162b2 and ChAdOx1 nCoV-19 Single dose: 48.7% (95% CI, 45.5–51.7). BNT162b2 Double dose: 93.7% (91.6–95.3) ChAdOx1 nCoV-19 Double dose: 74.5% (68.4–79.4)
[22]	Test negative case-control study	25,589 participants aged ≥18. January–July 2021	US	Χ	BNT162b2, mRNA-1273	Results aggregated against all variants *****: mRNA-1273: 86% (95% CI 81–90.6) *****; BNT162b2: 76% (69–81) ≥14 days after the second dose. Effectiveness against B.1.1.7 was estimated to be higher than B.1.617.2
[23]	Observational study	126,367 fully vaccinated individuals. February–April 2021	US	Χ	BNT162b2, mRNA-1273	High efficacy in fully vaccinated individuals. Did not quantify.
[24]	Observational case-control study	1210 hospitalized adults aged ≥18. 11 March–5 May 2021	US	Χ	BNT162b2 and mRNA-1273	BNT162b2: 84.3% (95% CI 74.6–90.3) * mRNA-1273 90.0% (82.0–94.4). Both Double doses
[25]	Test negative case-control study	156,930 adults aged ≥70. 8 December 2020–19 February 2021	UK	√	BNT162b2, ChAdOx1 nCoV-19	BNT162b2 Single dose: 61% (95% CI 51–69) ***** from 28 to 34 days after vaccination and then plateaued, Double dose: 89% (85–93) after 14 days from the second dose. ChAdOx1 nCoV-19 Single dose: 60% (41–73) from 28 to 34 days, increasing to 73% (27–90) from day 35
[26]	Observational study	Residents of Israel aged ≥16. 24 January–3 April 2021	Israel	√	BNT162b2	95.3% (95% CI 94·9–95·7) ***** from 7 days or longer after the second dose
[27]	Prospective cohort study	23,324 hospital staff aged ≥18. 7 December 2020–5 February 2021	UK	√	BNT162b2	Single dose: 70% (95% CI 55–85) ***** 21 days after the first dose Double dose: 85% (74–96) 7 days after two doses
[28]	Retrospective cohort study	245,226 Pescara resident aged ≥18. 2 January 2021–21 May 2021	Italy	√	BNT162b2, ChAdOx1 nCoV-19, mRNA-1273	BNT162b2 Single dose: 55% (95% CI 0.34–0.60) *****, Double dose: 98% (0.01–0.04) ChAdOx1 nCoV-19 Single dose: 95% (0.03–0.08) mRNA-1273 S Single dose: 93% (0.02–0.26)
[29]	Prospective cohort study	10,412 care home residents aged ≥65. 8 December 2020–15 March 2021	UK	√	BNT162b2, ChAdOx1 nCoV-19	56% (95% CI 19–76) at 28 to 34 days and 62% (23–81) ***** at 35 to 48 days after a single dose of ChAdOx1 or BNT162b2
[30]	Prospective cohort study	23,324 staff working in hospitals. 8 December 2020–5 February 2021	UK	Χ	BNT162b2	Single dose: 72% (95% CI 58–86) ***** 21 days after the first dose Double dose: 86% (76–97) 7 days after two doses
[31]	Test negative case-control study	4964 patients aged ≥65. 10 December 2020–31 May 2021	Europe	√	BNT162b2, ChAdOx1 nCoV-19	BNT162b2 Single dose: 61% (95% CI 39–75) *****, Double dose: 87% (74–93) ChAdOx1 nCoV-19 Single dose: 68% (39–83)
[32]	Case-control study (Questionnaire-based)	41,151 questionnaire respondants and 3644 controls. 14 February–3 May 2021	France	√	BNT162b2, mRNA-1273	Double dose: 86% (95% CI 81–90) 7 days after second dose. Reported efficacy for both mRNA vaccines aggregated
[33]	Test negative case-control study	324,033 residents aged ≥16. 14 December 2020–19 April 2021	Canada	√	BNT162b2, mRNA-1273	Single dose: 61% (95% CI 56–66) ≥14 days after the first dose Double dose: 90% (85–94) ≥7 days after the second dose (for both vaccines aggregated) Double dose against all variants combined: BNT162b2: 91% (88–93), mRNA-1273: 94% (86–97)
[34]	Observational study	25,558 health care workers aged 20–69. 15 March 2020–13 May 2021	Canada	Χ	BNT162b2, mRNA-1273	Results aggregated against all variants *****: Compared with community infection rates (for both vaccines aggregated): Single dose: 54.7% (95% CI 44.8–62.9); Double dose: 84.8% (75.2–90.7)
[35]	Test negative case-control study	16,993 participants aged ≥70. 4 April–1 May 2021	Canada	√	BNT162b2, mRNA-1273	Single dose: 67% (95% CI 57–75) (for both vaccines aggregated, 85% of the study population were given BNT162b2)
[36]	Test negative case-control study	682,071 symptomatic individuals aged ≥16. December 2020–March 2021	Canada	Χ	BNT162b2, ChAdOx1 nCoV-19, mRNA-1273	Single dose: mRNA-1273 83% (95% CI 80–86); BNT162b2 66% (64–68); ChAdOx1 64% (60–68) ≥14 days after the first dose Double dose: mRNA-1273: 92% (86–96); BNT162b2: 89% (86–91) ≥7 days after the second dose. Insufficient data for ChAdOx1

+ NVX-CoV2373 (Novavax), ChAdOx1 nCoV-19 (AZD1222/Oxford–AstraZeneca), BNT162b2 (Pfizer–BioNTech), mRNA-1273 (Moderna) * Estimated efficacy due to the variant being dominant at the time of study. √ indicates the study has been peer reviewed and Χ indicates the study has not been peer reviewed.

**Table 3 vaccines-09-01305-t003:** Vaccine Effectiveness against Beta (B.1.351) and Gamma (P.1) Variants.

Reference	Study Type	Population and Period	Location	Peer-Reviewed	Vaccine +	Reported Vaccine Efficacy
[18]	Phase 2 and Phase 3 trials	N/A	South Africa	Χ	NVX-CoV2373	Against B.1.351: 60% (CI Not reported) *****
[19]	Test negative case-control study	Qatari resident. February-March, 2021	Qatar	√	BNT162b2	Against B.1.351 Single dose: 16.9% (95% CI 10.4–23.0) Double dose: 75.0% (70.5–78.9) ≥14 days after the second dose
[20]	Test negative case-control study	104, 884 PCR-confirmed and negative for B.1.351 Qatari residents. 28 December 2020–10 May 2021	Qatar	√	mRNA-1273	Against B.1.351 Single dose: 61.3% (95% CI 56.5–65.5) ≥14 days after the first dose Double dose: 96.4% (91.9–98.7) ≥14 days after the second dose
[32]	Case-control study (Questionnaire-based)	41,151 questionnaire respondants and 3644 controls. 14 February–3 May 2021	France	√	BNT162b2, mRNA-1273	Reported efficacies against both B.1.351 and P.1 combined: Double dose: 77% (95% CI 63–86) 7 days after the second dose. Reported efficacy for both mRNA vaccines aggregated
[33]	Test negative case-control study	324,033 residents aged ≥16. 14 December 2020–19 April 2021	Canada	√	BNT162b2, mRNA-1273	Reported efficacies against both B.1.351 and P.1 combined: Single dose: 43% (95% CI 22–59) ≥14 days after the first dose Double dose: 88% (61–96) ≥7 days after the second dose (for both vaccines aggregated) Double dose against all variants combined: BNT162b2: 91% (88–93), mRNA-1273: 94% (86–97)
[34]	Observational study	25, 558 health care workers aged 20–69. 15 March 2020–13 May 2021	Canada	Χ	BNT162b2, mRNA-1273	Results aggregated against all variants *****: Compared with community infection rates (for both vaccines aggregated): Single dose: 54.7% (95% CI 44.8–62.9); Double dose: 84.8% (75.2–90.7)
[35]	Test negative case-control study	16, 993 participants aged ≥70. 4 April–1 May 2021	Canada	√	BNT162b2, mRNA-1273	Against P.1 Single dose: 61% (95% CI 45–72) (for both vaccines aggregated, 85% of study population were given BNT162b2)
[36]	Test negative case-control study	682,071 symptomatic individuals aged ≥16. December 2020–March 2021	Canada	Χ	BNT162b2, ChAdOx1 nCoV-19, mRNA-1273	Reported efficacies against both B.1.351 and P.1 combined: Single dose: mRNA-1273: 77% (95%CI 63–86); BNT162b2: 60% (52–67); ChAdOx1: 48% (28–63) ≥14 days after the first dose Double dose: BNT162b2: 84% (69–92) ≥7 days after the second dose; Insufficient data for ChAdOx1 and mRNA-1273
[37]	Phase 2a-b Randomized, observer-blinded, placebo-controlled trial	6324 participants, aged 18–84 without HIV and 18–64 with HIV. 17 August –25 November 2020	South Africa	√	NVX-CoV2373	Overall efficacy against B.1.351: 49.4% (95% CI 6.1–72.8) Post hoc vaccine efficacy against B.1.351: 51.0% (−0.6–76.2) among HIV-negative participants
[38]	Phase 3 randomized, double-blind, placebo-controlled trial	6576 participants from South Africa aged ≥18. 21 September 2020–22 January 2021	South Africa	√	Ad26.COV2.S	Against B.1.351 (Represented 95% of cases) Single dose: 52.0% (95% CI 30.3–67.4) ≥14 days after the first dose increasing to 64.0% (41.2–78.7) ≥28 days after the first dose
[39]	Randomized, placebo-controlled, observer-blind Phase 1/2/3 study	44,165 aged ≥16 and 2264 aged 12–15. 27 July–29 October 2020	South Africa	Χ	BNT162b2	Against B.1.351: 100% (95% CI 53.5–100.0) * >7 days after the second dose
[40]	Test negative case-control study	43,774 residents aged ≥70. 17 January–29 April 2021	Brazil	√	CoronaVac	Against P.1 ***** Single dose: 12.5% (95% CI 3.7–20.6) ≥14 days after the first dose Double dose: 46.8% (38.7–53.8) ≥14 days after the second dose
[41]	Test negative case-control study	106,329 health care workers aged ≥18. 19 January–25 March 2021	Brazil	Χ	CoronaVac	Against P.1 ***** Single dose: 49.6% (95% CI 11.3–71.4) ≥14 days after the first dose Double dose: 36.8% (-54.9–74.2) ≥14 days after the second dose
[42]	Double-blind, randomized, controlled trial	2026 HIV-negative adults aged 18–64. 24 June–9 November 2020	South Africa	√	ChAdOx1 nCoV-19	Overall efficacy against B.1.351: 21.9% (95% CI −49.9–59.8) >14 days after second dose Secondary-outcome analysis efficacy against B.1.351: 10.4% (−76.8–54.8) >14 days after the second dose
[43]	Retrospective cohort study	378 residents from long-term care facilities. 15 January–19 May 2021	France	Χ	BNT162b2	Against B.1.351 infection: 49% (95% CI 14–69) >7 days after the second dose Against B.1.351 severe COVID: 86% (67–94) >7 days after the second dose

+ NVX-CoV2373 (Novavax), Ad26.COV2.S (Johnson & Johnson/Janssen), ChAdOx1 nCoV-19 (AZD1222/Oxford–AstraZeneca), BNT162b2 (Pfizer–BioNTech), mRNA-1273 (Moderna). * Estimated efficacy due to the variant being dominant at the time of study. √ indicates the study has been peer reviewed and Χ indicates the study has not been peer reviewed.

**Table 4 vaccines-09-01305-t004:** Vaccine Effectiveness Against Delta (B.1.617.2) Variant.

Reference	Study Type	Population and Period	Location	Peer-Reviewed	Vaccine +	Reported Vaccine Efficacy
[21]	Test negative case-control study	19,109 participants aged ≥16. 26 October 2020–30 May 2021	UK	√	BNT162b2, ChAdOx1 nCoV-19	BNT162b2 and ChAdOx1 Single dose: 30.7% (95% CI, 25.2–35.7) BNT162b2 Double dose: 88.0% (85.3–90.1) ChAdOx1 nCoV-19 Double dose: 67.0% (61.3–71.8)
[22]	Test negative case-control study	25,589 participants aged ≥18. January–July 2021	US	Χ	BNT162b2, mRNA-1273	Results aggregated against all variants *: mRNA-1273: 86% (95% CI 81–90.6); BNT162b2: 76% (69–81) ≥14 days after the second dose. During July 2021 (high prevalence of B.1.617.2): mRNA-1273: 76% (58–87); BNT162b2: 42% (13–62)
[36]	Test negative case-control study	682,071 symptomatic individuals aged ≥16. December 2020–March 2021	Canada	Χ	BNT162b2, ChAdOx1 nCoV-19, mRNA-1273	Single dose: mRNA-1273: 72% (95% CI 57–82); BNT162b2: 56% (45–64); ChAdOx1: 67% (44–80) ≥14 days after first dose Double dose: BNT162b2: 87% (64–95) ≥7 days after the second dose; Insufficient data for ChAdOx1 and mRNA-1273
[44]	Observational Study	1945 reported cases. 27 April–6 June 2021	US	√	BNT162b2, mRNA-1273, Ad26.COV2.S	Against symptomatic infection for 2 weeks: 78% (95% CI 71–84) * for Mesa County (most cases were B.1.617.2), aggregated for all vaccines
[45]	Retrospective cohort study	3,436,957 participants aged ≥12. 14 December 2020–8 August 2021	US	Χ	BNT162b2	Against B.1.617.2: 75% (95% CI 71‒78) double dose Against other variants: 91% (88‒92) double dose
[46]	Prospective cohort study	4217 participants. 14 December 2020–14 August 2021	US	Χ	BNT162b2, mRNA-1273	B.1.617.2 predominant period: 66% (95% CI 26–84) ***** Before B.1.617.2 predominant period: 91% (81–96) For fully vaccinated participants, aggregated for both vaccines
[47]	Test negative case-control study	1,140,337 PCR-confirmed with Delta and PCR-negative Qatari residents. 23 March–21 July 2021	Qatar	Χ	BNT162b2, mRNA-1273	BNT162b2 Single dose: 64.2% (95% CI 38.1–80.1) BNT162b2 Double dose: 53.5% (43.9–61.4) mRNA-1273 Single dose: 79.0% (58.9–90.1) mRNA-1273 Double dose: 84.8% (75.9–90.8) ≥14 days after first dose and ≥14 days after second dose, respectively
[48]	Test negative case-control study	366 participants aged 18–59. 18 May–20 June 2021	China	√	CoronaVac, CNB	Double dose: 59.0% (95% CI 16.0–81.6) Results aggregated for all vaccines
[49]	Test negative case-control study	5143 participants. 1 April–31 May 2021	India	Χ	ChAdOx1 nCoV-19	Overall: 63.1% (95% CI 51.5–72.1) Only single dose: 46.2% (31.6–57.7)
[50]	Observational study	Population not specified. 16 January–28 June 2021	US	Χ	BNT162b2, mRNA-1273, Ad26.COV2.S	Estimated against B.1.617.2: 82% (95% CI 78–85) *****

+ CNB (China National Biotec inactivated vaccine), BNT162b2 (Pfizer–BioNTech), mRNA-1273 (Moderna), Ad26.COV2.S (Johnson & Johnson/Janssen), ChAdOx1 nCoV-19 (AZD1222/Oxford–AstraZeneca). * Estimated efficacy due to the variant being dominant at the time of study. √ indicates the study has been peer reviewed and Χ indicates the study has not been peer reviewed.

## Data Availability

Not applicable.

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
