# Peer review of "Vaccine versus Variants (3Vs): Are the COVID-19 Vaccines Effective against the Variants? A Systematic Review"

_vaccines, 2021, doi:10.3390/vaccines9111305_

Round 1

Reviewer 1 Report

Dear Editor,

The manuscript entitled: “Vaccine Versus Variants (3Vs): Are the COVID-19 Vaccines Effective Against the Variants? A Systematic Review” Hayawi K et al., describes the systematically review reporting vaccine effectiveness against SARS-CoV-2 variants. The results evidenced strong protection against the Alpha (B.1.1.7) variant. Furthermore, the results are not conclusive against the Beta (B.1.351) variant. Protection against Gamma (P.1) variant was lower, and finally, the data on Delta (B.1.617.2) variant is limited but indicates lower protection compared to other variants.  

I think this manuscript is current topic and can be worth publishing in the present form.

Author Response

We thank reviewer 1 for accepting our work, no concerns to address from his/her side.

Reviewer 2 Report

Major points

1 The quality evaluation of each study is needed in systematic review, but is not performed in this study.

2 In the discussion section, the authors just summarize findings of this study. The purpose of the discussion is to interpret and describe the significance of your findings in light of what was already known about the research problem being investigated, and to explain any new understanding or fresh insights about the problem after you've taken the findings into consideration. In this manuscript, little description like discussion is seen.

3 Page 11 Line 258-2854.2. Protection for Older Population Against COVID-19 Variants

 In this section, results of studies of reference [24, 34, 39] were described as the discussion for protection for older population against COVID-19 Variants. They are not discussion, and just description of results. Furthermore, for example, in the section “3.1. Vaccine Protection Against Alpha (B.1.1.7) SARS-CoV-2 Variant, no description is found concerning these studies, and suddenly emerged in this section. It is weird.

Minor points

1 Reference number should not be in the context.

For example, “[15] reported a drop in efficacy”.

2 “COVID-19 Variants” this term is not correct.

COVID-19 is a contagious disease caused by SARS-CoV-2. The term, “Variants” is used for virus or pathogens. Actually, the term “SARS-CoV-2 variants” is used in articles.

3 Page 12 Line 297-302

This part should be placed in the Discussion section.

4 Page 4 Line 137

“two doses” means “double doses”?

Author Response

Reviewer#2, Major Point # 1: The quality evaluation of each study is needed in systematic review but is not performed in this study.

Author response:  We would like to thank the reviewer for his/her feedback. We agree that the quality evaluation of each study is important. Therefore, we have focused on including papers from reputable peer-reviewed journals. However, due to the research topic being ongoing and active, we have also included pre-prints to highlight latest research findings. Nonetheless, non-peer-reviewed articles were highlighted appropriately.

Reviewer#2, Major Point # 2: In the discussion section, the authors just summarize findings of this study. The purpose of the discussion is to interpret and describe the significance of your findings in light of what was already known about the research problem being investigated, and to explain any new understanding or fresh insights about the problem after you've taken the findings into consideration. In this manuscript, little description like discussion is seen.

Author response:  We would like to thank the reviewer for his/her feedback. Based on the comment, we have further enriched the discussion section to reflect the findings in the context of previous research.

Reviewer#2, Major Point # 3: Page 11 Line 258-2854.2. Protection for Older Population Against COVID-19 Variants

In this section, results of studies of reference [24, 34, 39] were described as the discussion for protection for older population against COVID-19 Variants. They are not discussion, and just description of results. Furthermore, for example, in the section “3.1. Vaccine Protection Against Alpha (B.1.1.7) SARS-CoV-2 Variant, no description is found concerning these studies, and suddenly emerged in this section. It is weird.

Author response:  We would like to thank the reviewer for his/her feedback. The results section was kept concise to make it simple for readers to extract the necessary information. Moreover, we believe that vaccine efficacy against variants for older population is an important point of discussion and as such, discussion on references [24, 34, 39] which focused their study design on older population was necessary and relevant.

Reviewer#2, Minor Point # 1: Reference number should not be in the context.

For example, “[15] reported a drop in efficacy”.

Author response:  We would like to thank the reviewer for his/her feedback. We have made the necessary changes to reflect the reviewer’s comments.

Reviewer#2, Minor Point # 2: “COVID-19 Variants” this term is not correct.

COVID-19 is a contagious disease caused by SARS-CoV-2. The term, “Variants” is used for virus or pathogens. Actually, the term “SARS-CoV-2 variants” is used in articles.

Author response:  We would like to thank the reviewer for his/her feedback. We have made the necessary changes to reflect the reviewer’s comments.

Reviewer#2, Minor Point # 3: Page 12 Line 297-302

This part should be placed in the Discussion section.

Author response:  We would like to thank the reviewer for his/her feedback. We have made the necessary changes to reflect the reviewer’s comments.

Reviewer#2, Minor Point # 4: “two doses” means “double doses”?.

Author response:  We would like to thank the reviewer for his/her feedback. Yes, they are the same and we have replaced “double doses” with “two doses” for consistency.

Reviewer 3 Report

In the current manuscript the authors provide a systemic review of the literature concerning the efficacy of currently available COVID-19 vaccines against SARS-CoV-2 variants. The design of the authors' database search is well-explained and sound. Also, the authors' provide a helpful summary of the papers identified using their search criteria. However, the manuscript has a number of flaws:

  1. The presentation of vaccine efficacies (Table 1 and 2) is meaningless without providing key data- age range, study population, and the timeline of the study. These factors, as well as some others, can dramatically impact VE. Age is important since it has been reported that antibody responses and vaccine efficacy may be lower in individuals aged 65 yrs and above. Study population is key because factors such as underlying health conditions and risk for exposure can also influence VE. Timeline is critical since the dominance of certain variants has changed throughout the pandemic. Additionally, it's important to indicate the timepoint after vaccination VE was measured since a drop in VE may be due to waning immunity.
  2. Without these considerations, comparing one vaccine platform to another comes with several caveats. The authors fail to mention such caveats/limitations of their analyses in their Discussion/Conclusion.
  3. The author's statement that "not enough data is available for other variants besides alpha, beta, gamma, and delta" requires context. The primary reason for this is the dominance of other variants. Likewise, the number of respective studies available for alpha, beta, gamma, and delta are likely heavily influenced by the identity of the dominant variant circulating when a given study was performed. Perhaps this warrants an editorial comment by the authors that additional studies on the real world effectiveness of vaccines against other variants is needed. 

Author Response

Reviewer#3, Comment # 1: “The presentation of vaccine efficacies (Table 1 and 2) is meaningless without providing key data- age range, study population, and the timeline of the study. These factors, as well as some others, can dramatically impact VE. Age is important since it has been reported that antibody responses and vaccine efficacy may be lower in individuals aged 65 yrs and above. Study population is key because factors such as underlying health conditions and risk for exposure can also influence VE. Timeline is critical since the dominance of certain variants has changed throughout the pandemic. Additionally, it's important to indicate the timepoint after vaccination VE was measured since a drop in VE may be due to waning immunity”

Author response:  We would like to thank the reviewer for this valuable comment. We agree that study population and timelines are important and have updated the manuscript to reflect the changes.

Reviewer#3, Comment # 2: “Without these considerations, comparing one vaccine platform to another comes with several caveats. The authors fail to mention such caveats/limitations of their analyses in their Discussion/Conclusion”

Author response:  We would like to thank the reviewer for his/her feedback. Given that we have addressed the previous comment, the limitation does not exist any further.

Reviewer#3, Comment # 3: “The author's statement that "not enough data is available for other variants besides alpha, beta, gamma, and delta" requires context. The primary reason for this is the dominance of other variants. Likewise, the number of respective studies available for alpha, beta, gamma, and delta are likely heavily influenced by the identity of the dominant variant circulating when a given study was performed. Perhaps this warrants an editorial comment by the authors that additional studies on the real-world effectiveness of vaccines against other variants is needed”

Author response:  We would like to thank the reviewer for his/her feedback. We agree with the suggestion and have added this context to the manuscript.

Round 2

Reviewer 2 Report

Important points that I pointed out in the previous review report are not undertaken. Moreover, a new problem was found. The authors should address these problems sincerely.

Major points

1 The quality evaluation of each study is needed in systematic review, but is not performed in this study.

Quality assessment of each study that was selected in the review is an essential process for systematic review. After the screening process is complete, the systematic review team must assess each article for quality and bias. There are tools for quality assessment according to the type of article, including CASP, AMSTER, NOC, ROB, STROBE et al. Authors’ excuse is not to the point.

2 In the discussion section, the authors just summarize findings of this study.

Improvement is not sufficient. Most of descriptions are repetition of results. In the discussion, description about results should be kept to the minimum. You must interpret and describe the significance of your findings in light of what was already known about the research problem being investigated, and to explain any new understanding or fresh insights about the problem after you've taken the findings into consideration.

3 Page 11 Line 258-2854.2. Protection for Older Population Against COVID-19 Variants

Line267-291 These parts should be written in the Result section. The authors should describe how do they think these findings. Addition of known knowledge (line260-267) alone cannot make discussion.

References 24, 34, and 39 include only elderly subjects. But data addressing older people is included in other references. It cannot be unacceptable to neglect these studies.

4 3.1. Vaccine Protection Against Alpha (B.1.1.7) SARS-CoV-2 Variant

I cannot understand why data of reference 18-22 is written and others is not written. Since readers can find data in Tables, data (concrete figures) is, in general, not necessary in the text. Overall trend of studies reviewed should be written.

If findings are described by study design (case-control study, cohort study etc), readers can understand the contents.

5 Tables

The orders of the references should be improved. For example, in numerical order or by study design.

Minor points

1 Reference number should not be in the context.

Some parts that are applied to this problem are remained without correction. The authors should carefully correct.

2 Page 4 Line 137 “two doses” means “double doses”?

My indication means unification of technical term is preferable.

Author Response

Reviewer#2, Major Point # 1: The quality evaluation of each study is needed in systematic review but is not performed in this study.

Author response:  We would like to thank the reviewer for his/her feedback. We agree that the quality evaluation of each study is important in systematic review. Therefore, we have conducted quality evaluation using Risk Of Bias (ROB) approach proposed by (Hoy et al., 2012) and we reported the results of this evaluation in Table 1. ROB consists of assessing the quality of each study using 10 criteria/questions including for instance representation, data collection, case definition, test validity, and others. ROB score is calculated for each study considering the criteria stated above, and the final score is a sum of all negative answers. Scores of 0-3 are considered low risk, 4-6 are moderate risk, and 7-10 are high risk. The resulted score is in line with the Cochrane GRADE criteria of low, moderate, or high risk of bias. The full scoring details are provided as a supplementary material.

Reviewer#2, Major Point # 2: In the discussion section, the authors just summarize findings of this study. Most of descriptions are repetition of results. In the discussion, description about results should be kept to the minimum. You must interpret and describe the significance of your findings in light of what was already known about the research problem being investigated, and to explain any new understanding or fresh insights about the problem after you've taken the findings into consideration.

Author response:  We would like to thank the reviewer for his/her feedback. We have further improved the discussion section to highlight the strengths and limitations of the study. We highlighted the significance of our findings resulted from this study in the discussion section under sub-section 4.4. This provides a discussion on the implications of our study as well as possible future research directions. Although this systematic review is the first one to the best of our knowledge, we think from our analysis that it can be considered as new insights about vaccines efficacies. We also added a new subsection under discussion section 4 (Section 4.5) where we highlight the strengths and limitations of this review.

Reviewer#2, Major Point # 3: Page 11 Line 258-2854.2. Protection for Older Population Against COVID-19 Variants.

Line267-291 These parts should be written in the Result section. The authors should describe how do they think these findings. Addition of known knowledge (line260-267) alone cannot make discussion.

References 24, 34, and 39 include only elderly subjects. But data addressing older people is included in other references. It cannot be unacceptable to neglect these studies.

Author response: We would like to thank the reviewer for his/her feedback. Protection against older population is an important research question and we have linked our discussions in the context of existing research. The three references were selected because their study population solely comprises of older population whereas most of the other studies included adults of various age groups. It is expected that this part of the discussion will be beneficial to researchers looking specifically for vaccine efficacy in older population considering the new variants. Moreover, the contents from 267-291 were included in the discussion section because they are describing the study design for the references that focused on older aged population. In the results section, the study design for each study was tabulated but the description is kept concise. We believe if we describe each study design in detail would make it more difficult for readers to extract the necessary information.

Reviewer#2, Major Point # 4: 3.1. Vaccine Protection Against Alpha (B.1.1.7) SARS-CoV-2 Variant

I cannot understand why data of reference 18-22 is written and others is not written. Since readers can find data in Tables, data (concrete figures) is, in general, not necessary in the text. Overall trend of studies reviewed should be written.

Author response:  We would like to thank the reviewer for his/her feedback. The data for these references are written in the text to make specific points including the wide variability in the efficacies mentioned against B.1.1.7 variant and an efficacy comparison between B.1.1.7 and other variant. In overall, the text was kept concise as the tables summarize the results clearly.   

Reviewer#2, Major Point # 5: The orders of the references should be improved. For example, in numerical order or by study design.

Author response: We would like to thank the reviewer for his/her feedback. We agree with the recommendation, and we have reordered the table references by numerical order.

Reviewer#2, Minor Point # 1: Reference number should not be in the context.

Some parts that are applied to this problem are remained without correction. The authors should carefully correct.

Author response:  We would like to thank the reviewer for his/her feedback. We have made the necessary changes to reflect the reviewer’s comments.  

Reviewer 3 Report

I thank the authors for fully addressing all of my concerns. The current manuscript is much improved and I support its consideration for acceptance. The only comment I have on the revised manuscript is:

  1. line 57: correct to SARS-CoV-2 Variants of Concern in the US

Author Response

Reviewer#3, Comment # 1: line 57: correct to SARS-CoV-2 Variants of Concern in the US

Author response:  We would like to thank the reviewer for his/her feedback. We have corrected as per the reviewer’s suggestion.

Round 3

Reviewer 2 Report

Some improvement is found, however, problems which should be improved are still left unimproved. Discussion is no good. Repetition of results is not discussion, as repeatedly pointed out. And the extent of discussion is biased. Authors need to read other articles concerning your research.

#1 Page13 Line261-274

The discussion in this paragraph needs more improvement. At first, a term “reference variant” suddenly emerged. What does “reference variant” mean? In the original paper you cited used a tern “original variant”. And you used “original variant” in the paragraph. The unification of term is important because readers will be confused. I pointed out unification of terms before. “two doses” and “double doses. These should be unified. Results of effect of vaccines against reference variant is not present in the Results.

Second, “The vaccine efficacy for a single dose of Ad26.COV2.S in the US was reported to be 72.0% compared to 64.0% in South Africa, indicating lower protection against the beta variant compared to the reference variant.” From which reference does “72%” in US come? You should not compare data between other countries. Comparison must be done in the same population.

If you state that effect of vaccine against variants decreased, you need to discuss about it. Is the extent of this decline within allowance? Or not? Is the decrease greater compared with those seen in other virus variants? There are several points to discuss from data you reviewed. Readers require for the discussion regarding these points. Repetition of data shown in the tables will disappoint them.

#2 Line 284-308

Description of these data is meaningless. I pointed out in the previous reports twice. Readers can find these data in Tables without your description in the Discussion. You should describe the trend of the data. Moreover, I cannot find what you want to claim in this discussion. For example, what does the evidence that BNT162 showed an effect of prevention with more that 80% means? How is when compared with younger population? Does the known knowledge you cited apply to the data? It makes no sense without real discussion.

You should shorten the repetition of data to less than half and increase the discussion that you can think and consider from data with citing previous research. In all research articles, such discussion is done.

And authors do not answer the problem that older people are included other studies but they are not referred in the discussion.

#3 Evidence from articles reviewed in this study should be discussed in wide range. Discussion is confined to small range. For example, comparison between effect of vaccine against variants is more important than discussed in the present manuscript. Comparison in the same vaccine and in the same population, or making comparison totally.

#4 Overall, repetition of data is prevalent over the discussion section. Other academic articles include little in the discussion section and interpret and describe the significance of their findings in light of what was already known about the research problem being investigated, and to explain any new understanding or fresh insights about the problem after they have taken the findings into consideration.

#5 Line 323-325 Several other studies   only one study wad cited

Reference 53 is a preprint and has not been certified by peer review. It is not preferable to cite such article.  

Author Response

Reviewer#2, Major Point # 1: The discussion in this paragraph needs more improvement. At first, a term “reference variant” suddenly emerged. What does “reference variant” mean? In the original paper you cited used a tern “original variant”. And you used “original variant” in the paragraph. The unification of term is important because readers will be confused. I pointed out unification of terms before. “two doses” and “double doses. These should be unified. Results of effect of vaccines against reference variant is not present in the Results.

Second, “The vaccine efficacy for a single dose of Ad26.COV2.S in the US was reported to be 72.0% compared to 64.0% in South Africa, indicating lower protection against the beta variant compared to the reference variant.” From which reference does “72%” in US come? You should not compare data between other countries. Comparison must be done in the same population.

If you state that effect of vaccine against variants decreased, you need to discuss about it. Is the extent of this decline within allowance? Or not? Is the decrease greater compared with those seen in other virus variants? There are several points to discuss from data you reviewed. Readers require for the discussion regarding these points. Repetition of data shown in the tables will disappoint them.

Author response:  We would like to thank the reviewer for his/her feedback. We have unified the terms in the revised manuscript to only include ‘reference variant’. We have also removed the comparison of data across countries, and we focus only on comparing results within the same population. We have added further discussion regarding the decrease in efficacy of vaccine compared to influenza virus variants: “This decrease in vaccine efficacy is consistent with the influenza virus data where a reduction in vaccine efficacy against an emerging variant was reported in [51], with the effectiveness for influenza A/H1N1 of 65% being reduced to 39% for the circulating A/H3N2 variant.”

Reviewer#2, Major Point # 2: Description of these data is meaningless. I pointed out in the previous reports twice. Readers can find these data in Tables without your description in the Discussion. You should describe the trend of the data. Moreover, I cannot find what you want to claim in this discussion. For example, what does the evidence that BNT162 showed an effect of prevention with more that 80% means? How is when compared with younger population? Does the known knowledge you cited apply to the data? It makes no sense without real discussion.

You should shorten the repetition of data to less than half and increase the discussion that you can think and consider from data with citing previous research. In all research articles, such discussion is done.

And authors do not answer the problem that older people are included other studies but they are not referred in the discussion.

Author response: We would like to thank the reviewer for his/her feedback. We have shortened the repetition of data from this subsection. We concur that older population have been included in other studies but the studies in this subsection are exclusively performed on the older population. In other words, this subsection addresses only the older population (as the vulnerable population) and whether they are receiving substantial protection against the variants. With regards to the description of the trend of the data, we have discussed in the revised version of the paper under result section the data trends, and how its analysis has led to some interesting insights (e.g., Alpha variant efficacy is higher than other variants) and conclusion on vaccine efficacies assessment.

Reviewer#2, Major Point # 3: Evidence from articles reviewed in this study should be discussed in wide range. Discussion is confined to small range. For example, comparison between effect of vaccine against variants is more important than discussed in the present manuscript. Comparison in the same vaccine and in the same population, or making comparison totally.

Author response: We would like to thank the reviewer for his/her feedback. We have further elaborated on the discussion of vaccine against variants: “Given that vaccine effectiveness of higher than 50% could substantially reduce incidence of COVID-19 in vaccinated individuals [8], the existing vaccines from the available data indicates a good level of protection against the newer variants, especially with two doses. The efficacy against the alpha variant is reported to be the highest, followed by beta and gamma variants. However, from the limited data available, the efficacy against delta variant is indicated to be lower but still effective”

Reviewer#2, Major Point # 4: Overall, repetition of data is prevalent over the discussion section. Other academic articles include little in the discussion section and interpret and describe the significance of their findings in light of what was already known about the research problem being investigated, and to explain any new understanding or fresh insights about the problem after they have taken the findings into consideration.

Author response: We would like to thank the reviewer for his/her feedback. Based on the previous comments, we have reduced repetition of data and compared our findings with efficacy of vaccines on the influenza virus variants. We also extended our discussion to tackle the significance of our findings in light with other alternative works on vaccine’s efficacies evaluation. In addition, we highlighted the implications of this review work on various aspects including vaccine acceptance, increasing public health awareness among the population and address their concerns regarding the efficacies of the vaccines against new variants. Finally, we recognize some of the limitations of the review and we highlight how it can be addressed in future work.

Reviewer#2, Major Point # 5: Line 323-325 Several other studies only one study wad cited. Reference 53 is a preprint and has not been certified by peer review. It is not preferable to cite such article.

Author response: We would like to thank the reviewer for his/her feedback. We have removed the preprint citation and we rephrased the sentence accordingly: ‘Concerns over the vaccine efficacies against the new variants have been emphasized [54].’